# Towards hepatitis B elimination in Ghana: vaccination coverage and its predictors among informal sector workers in Kejetia, Kumasi, Ghana

Michael Agyemang Obeng[1]*, Daniel Kobina Okwan[2], Clinton Owusu Boateng[1], Godfred Yawson Scott[3], Joshua Kofi Attah[4], Senam Yawa Nunamey Ahadzie[1], Nathaniel Darko Antwi[1], Pius Takyi[1], De-Graft Kwaku Ofosu Boateng[5], Augustine Yeboah[1], Derrick Wedam[6], Emmanuella Asaamah Ofori[7], Richwonder Abla Ahiable[7], Ebenezer Oppong Gyamfi[1], Akwasi Amponsah Abrampah[8], Abu Abudu Rahamani[9]

1 Kumasi Centre for Collaborative Research in Tropical Medicine, Kwame Nkrumah University of Science and Technology, Kumasi, Ghana, 2 Department of Anatomy, Kwame Nkrumah University of Science and Technology, Kumasi, Ghana, 3 Department of Medical Diagnostics, Kwame Nkrumah University of Science and Technology, Kumasi, Ghana, 4 Department of Pathology, Komfo Anokye Teaching Hospital, Kumasi, Ghana, 5 Department of Pathology, Elbe Kliniken Stade-Buxtehude, Stade, Germany, 6 Department of Medical Laboratory Science, University of Energy and Natural Resources, Sunyani, Ghana, 7 Department of Laboratory Technology, Kumasi Technical University, Kumasi, Ghana, 8 Dawurampong Polyclinic, Ghana Health Service, Dawurampong, Ghana, 9 Department of Medical Laboratory Science, University for Development Studies, Tamale, Ghana

* michaelagyemangobeng@gmail.com

## Abstract

Hepatitis B virus (HBV) infection remains major public health concern in Ghana, where prevalence is high despite the availability of an effective vaccine. Informal sector workers represent a large proportion of the national workforce but have limited access to preventive health services. This study assessed hepatitis B vaccination coverage and its predictors among informal sector workers in Kejetia market, Kumasi, Ghana. A cross-sectional analytical study was conducted among 809 market workers selected using stratified random sampling across different occupational groups. Data were collected using structured interviewer-administered questionnaires covering sociodemographic characteristics, HBV awareness and knowledge, and vaccination history. Vaccination uptake was categorized as at least one dose (≥1) and full coverage (≥3 doses). Logistic regression analyses were used to identify predictors of vaccination uptake. Overall, only 137 (16.9%) had completed the three-dose schedule, though 256 (31.6%) of participants reported having received at least one dose of HBV vaccine. While 718 (88.8%) of respondents had heard of HBV infection, 619 (76.5%) reported very little or no knowledge about the disease, and 402 (49.7%) did not know its routes of transmission. Vaccination uptake was significantly higher among those who had received HBV-related health education, 175 (62.7%) compared to those who had not 18 (15.3%, p < 0.001). In the multivariate

**Data availability statement:** The data underlying the results presented in the study are available from Dryad at https://doi.org/10.5061/dryad.z8w9ghxsg.

**Funding:** The author(s) received no specific funding for this work.

**Competing interests:** The authors have declared that no competing interests exist.

logistic regression model, ≤ 30 years (aOR = 2.209, 95% CI [1.140–4.282], $p = 0.019$), secondary education (aOR = 2.426, 95% CI [1.220–4.824], $p = 0.012$), and tertiary education (aOR = 4.796, 95% CI [2.121–10.845], $p < 0.001$) were the independent predictors of vaccination status. HBV vaccination coverage among informal sector workers in Kejetia Market is alarmingly low despite high general awareness of the infection. Sociodemographic factors and receipt of HBV-related health education strongly influenced uptake. These findings underscore the urgent need for targeted health education and subsidized vaccination programs tailored to informal sector workers in Ghana to achieve national HBV elimination goals.

## Introduction

Hepatitis B virus (HBV) infection remains a significant global public health concern, despite the availability of a safe and highly effective vaccine [1–4]. In 2019, an estimated 296 million people were living with chronic HBV infection worldwide, resulting in approximately 820,000 deaths, primarily due to cirrhosis and hepatocellular carcinoma [1,2,5,6]. The African region bears a disproportionately high burden, with prevalence rates of 6–8%, while Ghana is classified as hyperendemic with prevalence estimates ranging from 8% to 15% [1,7].

Vaccination is the cornerstone of HBV prevention. The World Health Organization (WHO) recommends timely infant vaccination, completion of the vaccine series, and catch-up vaccination for unvaccinated adolescents and adults at risk [8,9]. However, in Ghana, despite the integration of HBV vaccination into the Expanded Programme on Immunization (EPI) since 2002, adult coverage remains low, with barriers including limited awareness, high costs, access challenges, and poor vaccine completion rates [10–13].

Informal sector workers are particularly vulnerable because of poor access to workplace health services, irregular income, and limited exposure to health education programs [14–16]. In Ghana, the informal sector represents a large proportion of the national workforce, yet HBV vaccination uptake within this group is poorly characterized [17,18]. The Kejetia Market in Kumasi, one of the largest open-air markets in West Africa, provides a unique setting to study this population. The market accommodates thousands of sellers, traders, and workers from diverse sociodemographic and cultural backgrounds, reflecting a microcosm of Ghana's informal economy [19]. Given their constant interaction with large numbers of people and limited access to preventive health interventions, market workers are at elevated risk for infectious diseases, including HBV [20].

Recent global policy shifts highlight the urgency of improving adult vaccination. In April 2022, the U.S. Advisory Committee on Immunization Practices (ACIP) recommended universal hepatitis B vaccination for all adults aged 19–59 years, removing the need for risk-based eligibility [21,22]. In March 2023, the U.S. Centers for Disease Control and Prevention (CDC) further recommended universal lifetime screening for HBV infection in adults aged ≥18 years using a triple serologic panel

[23]. These recommendations remove barriers associated with risk-based eligibility and hold promise for broader vaccine uptake and reduction in HBV cases.

Although these guidelines are U.S.-centric, they emphasize the global need to expand vaccine access to adults, particularly underserved groups. In line with Ghana's HBV elimination goals, it is essential to generate context-specific evidence on vaccination uptake and determinants among informal sector workers [24–26].

This study, therefore aimed to assess hepatitis B vaccination coverage, dose completion, and predictors among sellers and workers at Kejetia Market in Kumasi, Ghana, to inform targeted interventions for this high-risk and underserved population.

## Materials and methods

### Study design

This study employed a cross-sectional analytical design to assess the uptake of hepatitis B vaccination and its determinants among informal sector workers in Ghana. The study collected quantitative data from various informal sector workers to determine their vaccination status, knowledge, perceptions, and barriers to vaccination.

### Study setting

The study was conducted from 8th June 2024–5th January 2025 at Kejetia, a major commercial centre in Ghana, where a larger proportion of informal sector workers engage in trade, transport, and service-related activities. It is an open-air market place in Kumasi, a city in the Ashanti Region of Ghana, with over 8,000 stores and stalls [19]. This location was chosen due to its diverse informal workforce and potential risk of occupational exposure to HBV infection.

### Study population, inclusion and exclusion criteria

The study population was made up of randomly selected participants consisting of informal sector workers who belonged to any of the following occupational groups; butchers and meat sellers, fishmongers, tailors and seamstresses, head porters (kayayei), drivers and mates, food vendors, and general goods storekeepers. Participants were eligible if they were 18 years and above, had worked in Kejetia Market for at least six months, and provided informed consent to participate. Individuals who were mentally challenged were excluded.

### Sample size and sampling procedure

The minimum sample size was calculated using Cochran's formula for single proportions:

$$n = \frac{Z^2 \; x \; P(1-P)}{d^2}$$

Where:

$n$ = required sample size

$Z$ = standard normal deviation at 95% confidence level (1.96)

$p$ = assumed prevalence of hepatitis B vaccination uptake (50%), chosen to maximize sample size since no prior data were available for informal sector workers in Ghana

$d$ = margin of error (0.05)

Substituting:

$$n = \frac{(1.96)^2 \; x \; 0.5(1-0.5)}{(0.05)^2} = 384$$

To account for design effect due to stratified sampling, we doubled the sample size (384 × 2 = 768). An additional 5% was added to adjust for potential non-response (≈41). Thus, the final minimum required sample size was 809 participants, which was achieved in this study.

A stratified sampling technique was employed to ensure representation from different categories of informal sector workers. Participants were grouped into occupational strata (e.g., butchers, food vendors, drivers), and a proportionate random sampling approach was used within each stratum to obtain a representative sample. The sampling frame was developed through collaboration with market leadership and recognized trader associations, which maintain registers of traders and workers across the major market sections. Informal workers were first stratified into predefined subgroups based on the nature of their economic activities (e.g., butchers, food vendors, drivers).

The estimated population size of each subgroup was obtained from these records and validated through consultations with section heads and market executives. Proportional allocation was then applied to determine the number of participants to be selected from each stratum relative to its size within the overall market population. Within each stratum, eligible participants were selected using simple random sampling techniques (See Table 1.). This approach minimized selection bias and ensured that the final sample reflected the occupational composition of informal sector workers at Kejetia Market.

## Data collection

Data were collected using a structured, interviewer-administered questionnaire designed to capture knowledge and level of awareness of HBV infection, demographic data, occupational data, occupational hazards, HBV vaccination status, knowledge, perceptions, and barriers to vaccination. Trained research assistants conducted these face-to-face interviews with participants. Questionnaires were administered in English and local languages where necessary. Vaccination status was self-reported and, where possible, verified by vaccination cards.

Vaccination history was primarily based on participants' self-reporting due to the limited availability of vaccination cards in this informal population. To improve accuracy, participants were probed about the number of doses received, approximate dates, and sites of vaccination. Although recall bias cannot be excluded, self-reporting has been widely used in similar HBV epidemiological studies in Ghana and other African countries [11,27–30]. We also cross-checked available vaccination cards when presented by participants, which helped partially validate the self-reported information. The questionnaire used in this study is presented in the supplementary materials (S1 File).

**Table 1. Distribution of sampled participants across occupational subgroups among informal sector workers in Kejetia Market, Kumasi, Ghana (N = 809).**

| S/N | Occupational Group | Estimated Market Population | Sampled Participants |
|---|---|---|---|
| 1 | Butchers/meat sellers | 2,150 | 86 |
| 2 | Fishmongers | 1,090 | 44 |
| 3 | Tailors/seamstresses | 2,650 | 106 |
| 4 | Kayayei (market porters) | 1,880 | 77 |
| 5 | Drivers/mates | 730 | 30 |
| 6 | Food vendors | 3,400 | 136 |
| 7 | Storekeepers | 5,020 | 201 |
| 8 | Hawkers | 2,600 | 104 |
| 9 | Others (Barbers and hair dressers, security personnel, waste collectors and cleaners) | 620 | 25 |
| **Total** | | **20,140** | **809** |



### Study variables and measurements

The dependent variable was hepatitis B vaccination uptake, defined as receipt of at least one dose of the hepatitis B vaccine, as reported by the participant. For analytical purposes, vaccination uptake was coded as a binary variable (Yes/No). Participants who reported having received one or more doses of the hepatitis B vaccine were classified as having taken up vaccination, whereas those who reported never receiving any dose were classified as unvaccinated.

Information on completion of the full hepatitis B vaccination series (three or more doses) was collected and described separately but was not used as the primary outcome variable for the regression analysis.

The independent variables included socio-demographic factors (age, gender, educational level, etc), knowledge and perception of HBV and vaccination, barriers and facilitators to vaccination (cost, availability, employer support), and willingness to participate in an HBV education program.

### Definition of terms

**Hepatitis B vaccination coverage** refers to the proportion of the study participants who have received at least three shots of the HBV vaccine as recommended by the WHO.

**Incomplete vaccination** refers to individuals who have taken one or two shots of the HBV vaccine.

### Data analysis

Data were entered into Microsoft Excel 2019 and analyzed using SPSS version 26. Descriptive statistics (frequencies, percentages) were used to summarize participant characteristics. Chi-square tests were used to examine associations between independent variables and vaccination status. Logistic regression analysis was performed to identify independent predictors of vaccination uptake. Crude odds ratios (cOR) and adjusted odds ratios (aOR) with 95% confidence intervals (CI) were reported. A p-value $< 0.05$ was considered statistically significant.

### Ethical considerations

Ethical approval for this study was obtained from the Committee on Human Research Publication and Ethics (CHRPE), School of Medical Sciences, KNUST with approval number, CHRPE/AP 180/24. Permission was also obtained from Kejetia market management prior to data collection. All participants provided written informed consent. Data were anonymized to ensure confidentiality, and participation was voluntary with the right to withdraw at any stage without penalty. Written informed consent was obtained from all participants before recruitment.

## Results

### Sociodemographic and Behavioral Factors Associated with Hepatitis B Vaccination

Most of the participants were ≤ 30 years 322 (39.8%), were female 515 (63.7%), were married 408 (50.4%). The majority of them were Christians 590 (72.9%), from the Ashanti region 445 (55.0%), had basic education 357 (44.1%), and had stayed in urban areas for the greater part of their lives 610 (75.4%). Most of them did not have a close person living with hepatitis B 706 (87.3%). Majority of them stated that testing and vaccinating against Hepatitis B infection is expensive 362 (44.7%). Most of them were general goods storekeepers 201 (24.8%) (**Table 2**).

There was a significant association between age group ($p < 0.001$), marital status ($p < 0.001$), region ($p = 0.001$), educational level ($p = 0.009$), residence ($p = 0.014$), work type ($p = 0.016$) and vaccination status. However, gender ($p = 0.996$), religion ($p = 0.076$), having close person with hepatitis B infection ($p = 0.057$), perception about the cost of testing and vaccinating against Hepatitis B infection ($p = 0.617$) and Hepatitis B vaccination status were not statistically significant (**Table 2**).



**Table 2. Sociodemographic and Behavioral Factors Associated with Hepatitis B Vaccination.**

| Variables | Total (n = 809) | Vaccination Status | | p-value |
| --- | --- | --- | --- | --- |
| | | Not vaccinated (n = 553) | Vaccinated (n = 256) | |
| **Age Group** | | | | **< 0.001** |
| ≤ 30 years | 322(39.8) | 269(83.5) | 53(16.5) | |
| 31-40 years | 153(18.9) | 94(61.4) | 59(38.6) | |
| 41-50 years | 178(22.0) | 103(57.9) | 75(42.1) | |
| ≥ 51 years | 156(19.3) | 87(55.8) | 69(44.2) | |
| **Gender** | | | | 0.996 |
| Female | 515(63.7) | 352(68.3) | 163(31.7) | |
| Male | 294(36.3) | 201(68.4) | 93(31.6) | |
| **Marital Status** | | | | **< 0.001** |
| Co-Habiting | 97(12.0) | 80(82.5) | 17(17.5) | |
| Divorced | 24(3.0) | 19(79.2) | 5(20.8) | |
| Married | 408(50.4) | 237(58.1) | 171(41.9) | |
| Single | 247(30.5) | 196(79.4) | 51(20.6) | |
| Widowed | 33(4.1) | 21(63.6) | 12(36.4) | |
| **Religion** | | | | 0.076 |
| Christian | 590(72.9) | 390(66.1) | 200(33.9) | |
| Islamic | 201(24.8) | 150(74.6) | 51(25.4) | |
| Non-Religious | 18(2.2) | 13(72.2) | 5(27.8) | |
| **Region** | | | | **0.001** |
| Ashanti | 445(55.0) | 285(64.0) | 160(36.0) | |
| Northern | 149(18.4) | 121(81.2) | 28(18.8) | |
| Upper East | 70(8.7) | 44(62.9) | 26(37.1) | |
| Others | 145(17.9) | 103(71.0) | 42(29.0) | |
| **Educational Level** | | | | **0.009** |
| No Education | 81(10.0) | 61(75.3) | 20(24.7) | |
| Basic | 357(44.1) | 250(70.0) | 107(30.0) | |
| Secondary | 286(35.4) | 197(68.9) | 89(31.1) | |
| Tertiary | 85(10.5) | 45(52.9) | 40(47.1) | |
| **Residence** | | | | **0.014** |
| Rural | 199(24.6) | 150(75.4) | 49(24.6) | |
| Urban | 610(75.4) | 403(66.1) | 207(33.9) | |
| **Work Type** | | | | **0.016** |
| Butchers and meat sellers | 86(10.6) | 56(65.1) | 30(34.9) | |
| Drivers and mates | 30(3.7) | 25(83.3) | 5(16.7) | |
| Fish mongers | 44(5.4) | 28(63.6) | 16(34.6) | |
| Food vendors | 136(16.8) | 100(73.5) | 26(25.5) | |
| General goods storekeeper | 201(24.8) | 123(61.2) | 78(38.8) | |
| Market porters (Kayaye) | 77(9.5) | 60(77.9) | 17(22.1) | |
| Mobile workers (Hawkers) | 104(12.9) | 76(73.1) | 28(26.9) | |
| Tailors and seamstress | 106(13.1) | 65(61.3) | 41(38.7) | |
| Others | 25(13.1) | 20(80.0) | 5(20.0) | |
| **Do you have a close person who has Hepatitis B infection?** | | | | 0.057 |
| No | 706(87.3) | 491(69.5) | 215(30.5) | |
| Yes | 103(12.7) | 62(60.2) | 41(39.8) | |

*(Continued)*

 

**Table 2.** (Continued)

| Variables | Total (n = 809) | Vaccination Status Not vaccinated (n = 553) | Vaccinated (n = 256) | *p*-value |
|---|---|---|---|---|
| **How expensive is the testing and vaccination against Hepatitis B infection?** | | | | 0.617 |
| Cheap | 4(0.5) | 3(75.0) | 1(25.0) | |
| Expensive | 362(44.7) | 250(69.1) | 112(30.9) | |
| Reasonable | 347(42.9) | 240(69.2) | 107(30.8) | |
| Don't know | 96(11.9) | 60(62.5) | 36(37.5) | |

Data is presented as frequency (percentage). A *p* value of < 0.05 was considered statistically significant for hepatitis B vaccination status. The bold values indicate *p* values which are statistically significant for hepatitis B vaccination status.

## Hepatitis B vaccination coverage and influencing factors among the study participants

**Fig 1** illustrates the coverage of Hepatitis B vaccination among the study participants, along with the doses of the vaccine that they have taken.

Out of the total participants, most of them 553 (68.4%) had not been vaccinated against Hepatitis B infection, while 256 (31.6%) had received at least one dose of the Hepatitis B vaccine (**Fig 1A**). Only 137 (16.9%) of them had taken at least 3 doses of the Hepatitis B vaccine, while about 553 (68.4%) had not taken any dose (**Fig 1B**).

**Fig 2** illustrates Hepatitis B vaccination uptake and influencing factors among the participants.

Having prior knowledge about Hepatitis B infection was significantly associated with vaccine uptake (**Fig 2A**). Among those unvaccinated, the most common reason cited was the cost 326 (59.0%) (**Fig 2B**), which is further emphasized in (**Fig 2C**) where 440 (79.6%) of them stated that they would take the Hepatitis B vaccination if given to them for free. Among those vaccinated, 117 (45.7%) of them indicated that they took self-initiative to get vaccinated (**Fig 2D**).

## Awareness and knowledge of hepatitis B among participants

Awareness of Hepatitis B infection was generally high, with 718 (88.8%) indicating they were aware of the infection. When asked which group of people should be encouraged to take the hepatitis B vaccine, majority 507 (62.7%), correctly indicated it should be those who are not infected, however, 175 (21.6%) reported not knowing.

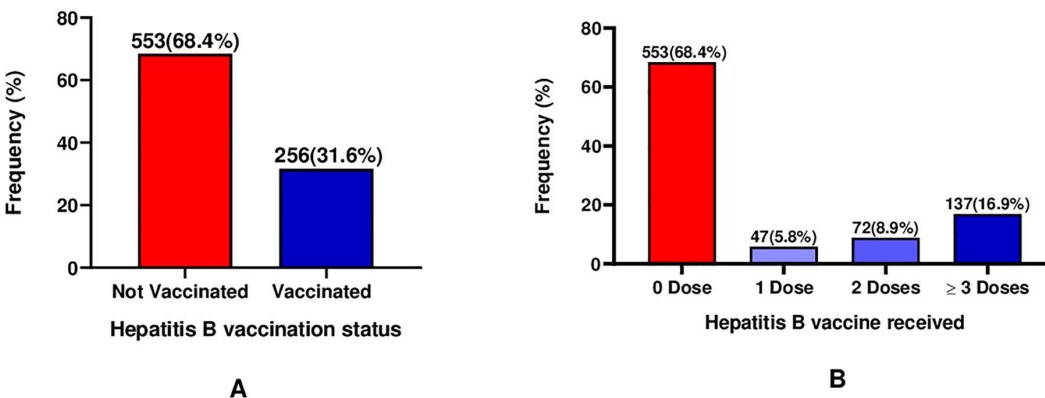

**Fig 1. Hepatitis B Vaccination Status and Dose Completion among Study Participants.**



**Fig 2. Hepatitis B vaccination uptake and influencing factors among the participants.**

A substantial 402 (49.7%) indicated they did not know the transmission route of Hepatitis B virus. Most 508 (62.8%) of them indicated that they were aware of vaccine availability against Hepatitis B infection. When asked about how much they know about Hepatitis B infection, most 323 (39.9%) of them indicated that they know very little about it (**Table 3**).

**Predictors of hepatitis B vaccination.** In the univariate logistic regression model, age group, marital status, region, educational level, and residence were predictors of hepatitis B vaccination. In the multivariate logistic regression model, respondents aged ≤ 30 years (aOR = 2.209, 95% CI [1.140–4.282], $p = 0.019$), had 2.21 times higher odds of being vaccinated against hepatitis B compared with those aged ≥ 51 years. Participants with secondary education (aOR = 2.426, 95% CI [1.220–4.824], $p = 0.012$), were approximately 2.4 times more likely to be vaccinated compared to those with no education. Similarly, respondents with tertiary education (aOR = 4.796, 95% CI [2.121–10.845], $p < 0.001$) had nearly five times higher odds of hepatitis B vaccination compared with those of no education (**Table 4**).

**Table 3. Awareness and Knowledge of Hepatitis B Among Participants.**

| Variables | Frequency (n = 809) | Percentage (%) |
|---|---|---|
| **Have you heard of Hepatitis B infection?** | | |
| No | 91 | 11.2 |
| Yes | 718 | 88.8 |
| **Which group of people are encouraged to take the Hepatitis B vaccine?** | | |
| Don't know | 175 | 21.6 |
| Those who are infected with Hepatitis B virus | 127 | 15.7 |
| Those who have not been infected with Hepatitis B virus | 507 | 62.7 |
| **How much do you know about Hepatitis B infection?** | | |
| A lot | 50 | 6.2 |
| Some | 140 | 17.3 |
| Very little | 323 | 39.9 |
| None | 296 | 36.6 |
| **Do you know if there is a vaccine available for Hepatitis B infection?** | | |
| No | 301 | 37.2 |
| Yes | 508 | 62.8 |
| **Which of the following are transmission routes of Hepatitis B virus?** | | |
| Mother-to-child | 105 | 13 |
| Physical contact | 119 | 14.7 |
| Through sex | 183 | 22.6 |
| Don't know | 402 | 49.7 |

Data is presented as frequency (percentage).

## Discussion

### Vaccination coverage among informal sector workers

This study assessed hepatitis B vaccination coverage and its predictors among informal sector workers at Kejetia market in Kumasi, Ghana. The overall vaccination coverage (≥3 doses) was 137 (16.9%) (Fig 1B), though 256 (31.6%) reported having taken at least one shot of the vaccine (≥1 dose). For adults, adequate protection against hepatitis B is obtained through a three-dose vaccination schedule [31]. This means that less than one in five of the study population is likely to remain protected against future HBV exposure. This low vaccination uptake is a serious public health challenge for a country with high HBV prevalence like Ghana and thus increased transmission risk [32]. These findings indicate critically low vaccination coverage, highlighting the urgent need for targeted public health interventions by policymakers to improve HBV vaccination coverage. These results highlight a significant gap in adult HBV prevention among this population, despite the availability of an effective vaccine. Findings from this study are consistent with the 18% vaccination coverage from Tanzania [29] and comparable to the 20.04% vaccination coverage (≥3 doses) reported from a systematic review among health care workers in Ethiopia [33]. Still, the HBV vaccination coverage recorded in this study is higher than the 11.4% reported among healthcare workers (HCWs) in Cameroon [34], 6.7% among HCWs in the Eastern Democratic Republic of the Congo [35] and the 5.4% among HCWs in Northwest Ethiopia [36]. Vaccination coverage in this study is lower than what has been reported among HCWs in Ghana (43.6%), HCWs in Nigeria (36.2%), HCWs in South Africa (49.0%), general adult population in Uganda (26.7%) and nurses Ethiopia (36.9%) [11,37–40]

The higher vaccination coverage reported among healthcare workers compared with the informal sector workers in this study may be explained by several factors. Healthcare workers often receive occupational health training that increases

**Table 4.  Predictors of Hepatitis B vaccination.**

| Variables | cOR | *p*-value | aOR | *p*-value |
| --- | --- | --- | --- | --- |
| **Age Group** | | | | |
| ≤ 30 years | 0.248(0.161-0.383) | **< 0.001** | 2.209(1.140-4.282) | **0.019** |
| 31-40 years | 0.791(0.503-1.246) | 0.312 | 0.881(0.486-1.595) | 0.675 |
| 41-50 years | 0.918(0.595-1.417) | 0.700 | 0.898(0.529-1.523) | 0.683 |
| ≥ 51 years | 1.0 | | | |
| **Marital Status** | | | | |
| Co-Habiting | 0.372(0.154-0.898) | **0.028** | 1.631(0.544-4.889) | 0.383 |
| Divorced | 0.461(0.137-1.550) | 0.211 | 2.235(0.561-8.912) | 0.254 |
| Married | 1.263(0.605-2.636) | 0.535 | 0.622(0.256-1.513) | 0.295 |
| Single | 0.455(0.210-0.987) | **0.046** | 1.243(0.448-3.447) | 0.676 |
| Widowed | 1.0 | | | |
| **Region** | | | | |
| Ashanti | 1.377(0.916-2.070) | 0.124 | 0.789(0.477-1.303) | 0.354 |
| Northern | 0.567(0.329-0.979) | **0.042** | 1.183(0.592-2.363) | 0.635 |
| Upper East | 1.449(0.793-2.649) | 0.228 | 0.571(0.260-1.252) | 0.162 |
| Others | 1.0 | | | |
| **Educational Level** | | | | |
| No Education | 1.0 | | | |
| Basic | 1.305(0.751-2.270) | 0.345 | 1.282(0.673-2.443) | 0.449 |
| Secondary | 0.508(0.310-0.833) | **0.007** | 2.426(1.220-4.824) | **0.012** |
| Tertiary | 2.711(1.400-5.249) | **0.003** | 4.796(2.121-10.845) | **< 0.001** |
| **Residence** | | | | |
| Rural | 0.636(0.442-0.915) | **0.015** | 0.988(0.627-1.556) | 0.957 |
| Urban | 1.0 | | | |
| **Work Type** | | | | |
| Butchers and meat sellers | 2.143(0.731-6.283) | 0.165 | – | – |
| Drivers and mates | 0.800(0.203-3.155) | 0.750 | – | – |
| Fish mongers | 2.286(0.719-7.266) | 0.161 | – | – |
| Food vendors | 1.440(0.503-4.121) | 0.497 | – | – |
| General goods store keeper | 2.537(0.914-7.036) | 0.074 | – | – |
| Market porters (Kayaye) | 1.133(0.370-3.467) | 0.826 | – | – |
| Mobile workers (Hawkers) | 1.474(0.505-4.303) | 0.478 | – | – |
| Tailors and seamstress | 2.523(0.878-7.247) | 0.086 | – | – |
| Others | 1.0 | | | |

1.00, reference. Binary logistic regression analysis was performed to obtain odds ratios. A *p* value of <0.05 was considered statistically significant for hepatitis B vaccination status. The bold values indicate *p* values which are statistically significant for hepatitis B vaccination status. Abbreviations: aOR, adjusted odds ratio; CI, confidence interval; cOR, crude odds ratio.

awareness of blood-borne infections and the importance of vaccination. In many healthcare facilities, hepatitis B vaccination is recommended or facilitated through workplace policies, occupational health programs, or infection prevention initiatives. In contrast, informal sector workers typically lack structured workplace health programs and may face financial and logistical barriers to accessing vaccination services. These structural differences likely contribute to the lower vaccination coverage observed in informal sector populations.

## Hepatitis B vaccination uptake and influencing factors among the participants

The study revealed that age group, marital status, education level, residence and work type are the statistically significant variables influencing HBV vaccination uptake (Table 1). Our finding aligns with other African studies, though different populations, a similar trend exists. For example, among Ghanaian health facility staff, age group, marital status, and profession were all significantly associated with vaccination (p < 0.001) [11]. In Kampala, Uganda, lower education, age and marital status predicted greater vaccine hesitancy [41]. These concordant findings underscore the consistent influence of sociodemographic factors on adult HBV vaccine uptake across diverse African settings [40,42,43]. Vaccination coverage increased with increasing age group and increasing educational level. This study demonstrated that participants who had lived more in urban centers having higher odds of HBV vaccine coverage than their rural counterparts. This could be due to increased health educational programs and access to health facilities among urban dwellers compared to rural dwellers.

Married participants had significantly higher vaccination coverage compared to singles or those cohabiting. This may reflect the role of spousal and family influence in health-seeking behaviors, where partners encourage one another to pursue preventive health measures. Married individuals are also more likely to access healthcare services during antenatal or family planning visits, creating opportunities for HBV screening and vaccination [44]. Conversely, singles or cohabiting individuals may perceive themselves at lower risk or may lack the social support and motivation to prioritize preventive measures such as vaccination [45]. These findings indicate that family and social dynamics are important considerations in designing HBV vaccination campaigns.

Participants who had received hepatitis B–related health education demonstrated substantially higher vaccination uptake compared with those without prior health education, highlighting the importance of targeted health education in improving vaccine coverage (Fig 2A). The difference was statistically significant consistent with other studies [13,40,46]. Our study emphasizes that, public education can be one of the key ways to increase the vaccination coverage and encourage HBV preventive practices which will contribute to the 2030 HBV elimination goal as set by WHO.

Cost-related barriers emerged as the dominant reason for non-vaccination among unvaccinated participants, alongside time constraints, lack of specific reasons, and limited awareness of the hepatitis B vaccine. Notably, a substantial majority of unvaccinated participants indicated willingness to receive the vaccine if it were provided free of charge, underscoring the critical role of affordability in hepatitis B vaccination uptake (Fig 2C). Our findings show that, subsidizing the HBV vaccination and or probably making it free of charge to be covered under the national health insurance scheme can improve vaccination coverage among adults in Ghana.

## Awareness and knowledge of Hepatitis B among participants

While 717 (88.8%) cited having heard of HBV infection before, 619 (76.5%) reported either very little knowledge or no knowledge about hepatitis B infection. Again, when asked the route of transmission of HBV, almost half of them 402 (49.7%) said they do not know (Table 2). This pattern – high general awareness but low specific knowledge – has been observed in multiple Ghanaian studies. For example, barbers in Obuasi reported high awareness (~90%) yet poor understanding of transmission and prevention [20]. Similarly, a community study of young adults found widespread awareness but low knowledge and low testing/vaccination uptake [47], while a mixed-methods survey of street beauticians and barbers reported ~88% awareness but two-thirds with poor knowledge [48]. Low knowledge has also been linked with reduced testing and preventive behaviors in national surveys, underscoring the need for targeted education interventions among market populations [49].

## Strengths and limitations

Vaccination history was mainly based on participant recall due to limited availability of vaccination cards, which may have introduced recall bias. The proportion of participants who presented vaccination cards were 224 (82.1%) of those



who reported having taken the shot. This reduces the chances of recall bias. Also, factors such as previous HBV testing, perceptions of stigma, or cultural beliefs around vaccination were not deeply explored, which might also influence uptake. Other limitations include the absence of serologic testing to confirm vaccine-induced immunity (e.g., anti-HBs antibody levels), which limits our ability to distinguish between vaccination and actual immunologic protection. Also, the single-site nature of the study conducted at Kejetia Market in Kumasi, which may limit the generalizability of findings to other informal sector populations or geographic settings in Ghana. Notwithstanding, our study has several strengths. This is the first study, to our knowledge, that specifically assessed hepatitis B vaccination coverage and its determinants among informal sector workers in Kejetia market, one of the largest open-air markets in West Africa. The findings therefore provide context-specific evidence for a high-risk but under-researched population. Again, with 809 participants across diverse occupational groups (e.g., traders, food vendors, porters, butchers, seamstresses), the study achieved broad representation of informal workers, enhancing the generalizability of the findings to similar market settings in Ghana. Lastly, the study provides timely evidence to inform Ghana's HBV elimination efforts and highlights actionable areas such as health education and subsidized vaccination, which can guide interventions within informal sector populations.

## Conclusion

This study demonstrated that hepatitis B vaccination coverage among informal sector workers in Kejetia market, Kumasi, remains alarmingly low, with fewer than one in five participants completing the recommended three-dose schedule. Despite high awareness of hepatitis B, detailed knowledge about the infection and its transmission was poor, and uptake was strongly influenced by sociodemographic factors such as age, marital status, education level, residence, and work type. Importantly, receipt of HBV-related health education was associated with markedly higher vaccine uptake, underscoring the value of targeted educational interventions.

The findings highlight the urgent need to scale up accessible and affordable HBV vaccination programs tailored to informal sector workers, including integration into workplace and market-based health outreach initiatives. Policies such as subsidizing or covering vaccination under the National Health Insurance Scheme could address financial barriers, while structured health education campaigns may help close persistent knowledge gaps. Strengthening these efforts will be critical for Ghana to meet its HBV elimination targets and to protect vulnerable populations within the informal economy.

## Supporting information

**S1 File. Appendix.**
(DOCX)

## Acknowledgments

We express our greatest appreciation to the management of Kejetia market and all the study participants who willingly took part in this study.

## Author contributions

**Conceptualization:** Michael Agyemang Obeng, Daniel Kobina Okwan, De-Graft Kwaku Ofosu Boateng.

**Data curation:** Michael Agyemang Obeng, Daniel Kobina Okwan, Clinton Owusu Boateng, Godfred Yawson Scott, Joshua Kofi Attah, Senam Yawa Nunamey Ahadzie, Nathaniel Darko Antwi, Pius Takyi, Augustine Yeboah, Derrick Wedam, Emmanuella Asaamah Ofori, Richwonder Abla Ahiable, Ebenezer Oppong Gyamfi.

**Formal analysis:** Michael Agyemang Obeng, Daniel Kobina Okwan, Godfred Yawson Scott, Joshua Kofi Attah, Senam Yawa Nunamey Ahadzie, De-Graft Kwaku Ofosu Boateng, Augustine Yeboah, Ebenezer Oppong Gyamfi.

**Investigation:** Michael Agyemang Obeng, Daniel Kobina Okwan, Clinton Owusu Boateng, Godfred Yawson Scott, Joshua Kofi Attah, Senam Yawa Nunamey Ahadzie, Nathaniel Darko Antwi, Pius Takyi, Derrick Wedam, Emmanuella Asaamah Ofori, Richwonder Abla Ahiable, Ebenezer Oppong Gyamfi, Akwasi Amponsah Abrampah.

**Methodology:** Michael Agyemang Obeng, Daniel Kobina Okwan, Joshua Kofi Attah, Akwasi Amponsah Abrampah, Abu Abudu Rahamani.

**Project administration:** Michael Agyemang Obeng, Daniel Kobina Okwan, Clinton Owusu Boateng, Joshua Kofi Attah, Senam Yawa Nunamey Ahadzie, Pius Takyi, Abu Abudu Rahamani.

**Resources:** Michael Agyemang Obeng, Daniel Kobina Okwan, Senam Yawa Nunamey Ahadzie, De-Graft Kwaku Ofosu Boateng, Akwasi Amponsah Abrampah.

**Software:** Daniel Kobina Okwan.

**Supervision:** Michael Agyemang Obeng, Clinton Owusu Boateng, Joshua Kofi Attah, Senam Yawa Nunamey Ahadzie, De-Graft Kwaku Ofosu Boateng, Abu Abudu Rahamani.

**Validation:** Michael Agyemang Obeng, Daniel Kobina Okwan, Clinton Owusu Boateng, Godfred Yawson Scott, Senam Yawa Nunamey Ahadzie, Pius Takyi, De-Graft Kwaku Ofosu Boateng, Ebenezer Oppong Gyamfi, Abu Abudu Rahamani.

**Visualization:** Michael Agyemang Obeng, Daniel Kobina Okwan, Clinton Owusu Boateng, Godfred Yawson Scott, Joshua Kofi Attah, Senam Yawa Nunamey Ahadzie, Pius Takyi, De-Graft Kwaku Ofosu Boateng, Augustine Yeboah, Abu Abudu Rahamani.

**Writing – original draft:** Michael Agyemang Obeng, Daniel Kobina Okwan, Clinton Owusu Boateng, Godfred Yawson Scott, Joshua Kofi Attah, Senam Yawa Nunamey Ahadzie, Nathaniel Darko Antwi.

**Writing – review & editing:** Michael Agyemang Obeng, Daniel Kobina Okwan, Clinton Owusu Boateng, Godfred Yawson Scott, Joshua Kofi Attah, Senam Yawa Nunamey Ahadzie, Nathaniel Darko Antwi, Pius Takyi, De-Graft Kwaku Ofosu Boateng, Augustine Yeboah, Derrick Wedam, Emmanuella Asaamah Ofori, Richwonder Abla Ahiable, Ebenezer Oppong Gyamfi, Akwasi Amponsah Abrampah, Abu Abudu Rahamani.

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
