## [Decision Letter · Decision Letter 0]

11 Dec 2025

Dear Dr. Obeng,

Thank you for submitting your manuscript to PLOS ONE. After careful consideration, we feel that it has merit but does not fully meet PLOS ONE’s publication criteria as it currently stands. Therefore, we invite you to submit a revised version of the manuscript that addresses the points raised during the review process.

We look forward to receiving your revised manuscript.

Kind regards,

Jason T. Blackard, PhD

Academic Editor

PLOS One

[Committee on Human Research Publication and Ethics (CHRPE), School of Medical Sciences, KNUST].

Additional Editor Comments:

This is a cross-sectional analysis of HBV vaccination conducted in Kumasi, Ghana.

The epidemiologic and statistical methods are  standard and well described.  The results

Limitations should include the lack of serologic testing.  HBV vaccine-induced antibodies could be measured but were not.  Moreover, the study was a single site analysis; thus, generalizability to other at-risk populations or other locations may be low.

Reviewers' comments:

Reviewer's Responses to Questions

**Comments to the Author**

1. Is the manuscript technically sound, and do the data support the conclusions?

Reviewer #1: Partly

Reviewer #2: Partly

2. Has the statistical analysis been performed appropriately and rigorously?

Reviewer #1: Yes

Reviewer #2: Yes

3. Have the authors made all data underlying the findings in their manuscript fully available?

Reviewer #1: No

Reviewer #2: Yes

4. Is the manuscript presented in an intelligible fashion and written in standard English?

Reviewer #1: Yes

Reviewer #2: Yes

Reviewer #1: This manuscript provides valuable information on Hepatitis B vaccination uptake in a typically understudied population however, the authors could consider addressing the following concerns to improve on the manuscript.

Major

1. Study design - Carrying out logistic regression analysis converts the study from a descriptive one to analytical one.

2. Sampling technique - The authors could consider providing more details about how they obtained the sampling frame for the different subgroups (of informal workers at the Kejetia Market) that were used for the proportionate stratified random sampling

3.Study variables - in the definition of the dependent variable, it is unclear whether uptake of vaccination was referring to any number of doses or the full series (≥3doses) - Line 172-173

Minor

4. Findings - The authors are encouraged to check and correct errors in the cell entries (including percentages). For instance, in Table 1 (line 219), the percentages for cohabiting, having a close person with Hep B and the reasonability option in assessing how expensive testing and vaccination are have errors in the percentages.

Additionally More details/depth on the results of the multivariate logistic would be helpful to understand what the aOR mean.(lines264 -268)

5. Discussion - The discussion would be easier to follow if the participant groups being compared to the current study are stated and similar to the current study's participants (lines 311-315)

- It is also unclear if line 326 is referring to the current study, which will mean there is no need for the cited reference).

- consider editing Line 327 for clarity

6. The authors are encouraged to check all references to ensure that they are complete, correct and well-formatted according to the journal's requirement.

Additionally, Lines 339 to 340 and 346-350. The authors could consider summarising these as they are currently repetitions of the results presented in the figures.

Reviewer #2: Major issues

1. The manuscript addresses issues on the elimination of hepatitis B virus infections in Ghana by understanding the vaccination coverage and its predictors among informal sector workers in Kejetia, Kumasi, Ghana. The study has been well conducted and reported but I still have some concerns with the self-reporting of vaccination. Most of the papers cited on self-reporting had to do with healthcare workers so it may be understandable when that is used among them. But with a big proportion of participants having basic or no education, it may be necessary to justify that among this group this kind of self-reporting is reliable. There is therefore the need to address this issue even though it has been admitted as a limitation. The analysis showing that the self-reporting was reliable based on those with vaccination cards must be explained. How many people had cards? Perhaps, if that is reported it will give more strength to the study.

2. On page 18, the discussions there must be placed in context. The study population was not healthcare workers. The use of references suggesting trends like healthcare workers must be done with caution. An example, “Our finding aligns with other African studies. For example, among Ghanaian health facility staff, age group, marital status, and profession were all significantly associated with vaccination (p < 0.001) [11]. In Kampala, Uganda, lower education, age and marital status predicted greater vaccine hesitancy [41].” It must be clear in the discussion that though they are two different populations, a similar trend exists.

3. All numbers and percentages must be written as 515 (63.7%); and there must be a space between the number and the parenthesis. It is important that whenever a percentage is written, the number must precede it.

4. The authors must use data from the results sparing in the discussion. For example, “Among the unvaccinated, when asked the reason for not vaccinating, 69.0% cited cost of HBV screening and vaccination, 22.6% mentioned they do not have any reason, 16.0% cited busy schedule and 3.4% mentioned they do not know about the vaccine (Figure 2B). Meanwhile, when asked if one will take the vaccine if given free of charge, nearly 80% (79.6%) responded “Yes” and about one in five (20.4%) responded “No” (Figure 2C). “ All these could have been expressed succinctly to make the point that vaccine coverage may improve if offered freely.

5. The authors have a few typos and grammatical issues to address.

.

Reviewer #1: No

Reviewer #2: **Yes:** Kwamena SagoeKwamena SagoeKwamena SagoeKwamena Sagoe

---

## [Author Response · Author response to Decision Letter 1]

28 Jan 2026

RESPONSE TO REVIEWERS

Manuscript ID: PONE-D-25-51851

Title: Towards hepatitis B elimination in Ghana: vaccination coverage and its predictors among informal sector workers in Kejetia, Kumasi, Ghana

We sincerely thank the Academic Editor and the reviewers for the constructive feedback and for the opportunity to revise our manuscript. We have carefully addressed all comments and journal requirements. Our responses are detailed below, and corresponding revisions have been made in the revised manuscript (tracked and clean [unmarked] versions).

Comments from academic editor

Response: Thank you for this reminder. The manuscript has been reviewed and revised to conform fully with PLOS ONE formatting and style requirements, including file naming conventions, structure, and layout. The revised submission includes: a marked-up manuscript with tracked changes and a clean, unmarked version of the revised manuscript.

[Committee on Human Research Publication and Ethics (CHRPE), School of Medical Sciences, KNUST].

Response: The authors received no specific funding for this work.

Response: Not applicable

Response: No authors received a salary from any funder in relation to this study.

Response: These statements have been clearly included in the cover letter as requested.

Response: The ethics statement has been reviewed and confirmed to appear only in the Methods section of the manuscript in compliance with PLOS ONE guidelines.

Response: No reviewer recommended the citation of specific previously published works. Therefore, no additional references were required or added to the manuscript in response to this instruction.

Additional Editor Comments:

This is a cross-sectional analysis of HBV vaccination conducted in Kumasi, Ghana. The epidemiologic and statistical methods are standard and well described.

Response: We thank the Academic Editor for this positive assessment of the study design and analytical methods. No changes were required in response to this comment.

The results Limitations should include the lack of serologic testing. HBV vaccine-induced antibodies could be measured but were not. Moreover, the study was a single site analysis; thus, generalizability to other at-risk populations or other locations may be low.

Response: We appreciate this important suggestion. We have revised the Limitations subsection to explicitly acknowledge:

1. The absence of serologic testing to confirm vaccine-induced immunity (e.g., anti-HBs antibody levels), which limits our ability to distinguish between vaccination and actual immunologic protection.

2. The single-site nature of the study conducted at Kejetia Market in Kumasi, which may limit the generalizability of findings to other informal sector populations or geographic settings in Ghana.

These limitations are now clearly discussed to strengthen the transparency and interpretability of the study findings (See page 22, lines 396 - 400 of the unmarked manuscript).

Comments from Reviewer # 1

Reviewer #1: This manuscript provides valuable information on Hepatitis B vaccination uptake in a typically understudied population however; the authors could consider addressing the following concerns to improve on the manuscript.

Response: Thank you for the thoughtful and constructive review of our manuscript and for recognizing the value of the study in addressing hepatitis B vaccination uptake among an understudied informal sector population. We have carefully considered all comments and revised the manuscript accordingly. Our point-by-point responses are provided below.

Major

1. Study design - Carrying out logistic regression analysis converts the study from a descriptive one to analytical one.

Response: Thank you for this important methodological clarification. We agree with this observation and have revised the manuscript to clearly describe the study as a cross-sectional analytical study, rather than purely descriptive. This change has been reflected in the Abstract, Methods section, and Study Design description to ensure consistency with the analytical approach used, including logistic regression analysis (See page 3, line 42; page 7, line 120).

2. Sampling technique - The authors could consider providing more details about how they obtained the sampling frame for the different subgroups (of informal workers at the Kejetia Market) that were used for the proportionate stratified random sampling

Response: We appreciate this suggestion and have expanded the Sampling Technique subsection to provide additional detail. Specifically, we now clarify how the sampling frame for the various informal worker subgroups at Kejetia Market was obtained, including the identification of trader and worker categories through market association registers and leadership structures. We also explain how proportional allocation was applied across strata prior to random selection of participants (See pages 8,9, lines 154-164 of the unmarked manuscript).

3.Study variables - in the definition of the dependent variable, it is unclear whether uptake of vaccination was referring to any number of doses or the full series (≥3doses) - Line 172-173

Response: Thank you for highlighting this lack of clarity. We have revised the definition of the dependent variable to explicitly state that hepatitis B vaccination uptake refers to receipt of at least one dose of the hepatitis B vaccine, in line with the study objective. Where relevant, we also clarify distinctions between partial and complete vaccination (≥3 doses). The revised wording has been incorporated into the Study Variables section (See page 10, lines 182-189 of the unmarked manuscript).

Minor

4. Findings - The authors are encouraged to check and correct errors in the cell entries (including percentages). For instance, in Table 1 (line 219), the percentages for cohabiting, having a close person with Hep B and the reasonability option in assessing how expensive testing and vaccination are have errors in the percentages.

Additionally More details/depth on the results of the multivariate logistic would be helpful to understand what the aOR mean.(lines264 -268)

Response: We thank the reviewer for this careful observation. All tables have been thoroughly reviewed, and errors in cell entries and percentages in Table 1 (including cohabitation status, having a close person with hepatitis B, and perceived cost of testing/vaccination) have been corrected.

In addition, we have expanded the Results section to provide clearer interpretation of the logistic regression findings (See pages 15, 16, lines 282-288 of the unmarked manuscript).

5. Discussion - The discussion would be easier to follow if the participant groups being compared to the current study are stated and similar to the current study's participants (lines 311-315)

Response: We appreciate these helpful suggestions. The Discussion section has been revised to:

Clearly state the participant characteristics of studies being compared (e.g., healthcare workers, traders, adults in community settings), ensuring appropriate contextual alignment with the current study population (See page 19, lines 333-335 of the unmarked manuscript).

- It is also unclear if line 326 is referring to the current study, which will mean there is no need for the cited reference).

Response: Thank you. It is referring to the current study. This has been corrected accordingly (see page 19, line 346 of the unmarked manuscript).

- consider editing Line 327 for clarity

Response: This has been rectified accordingly (see page 19, lines 346, 347 of the unmarked manuscript)

6. The authors are encouraged to check all references to ensure that they are complete, correct and well-formatted according to the journal's requirement.

Additionally, Lines 339 to 340 and 346-350. The authors could consider summarising these as they are currently repetitions of the results presented in the figures.

Response: We thank the reviewer for this guidance. All references have been carefully reviewed to ensure they are complete, accurate, and formatted according to PLOS ONE requirements.

Additionally, we have revised lines 339–340 and 346–350 to summarize key findings rather than restating results already presented in figures. This has reduced redundancy and improved the conciseness of the manuscript (See page 20, lines 359-361, 367-373 of the unmarked manuscript).

Comments from Reviewer # 2

Reviewer #2: Major issues

1. The manuscript addresses issues on the elimination of hepatitis B virus infections in Ghana by understanding the vaccination coverage and its predictors among informal sector workers in Kejetia, Kumasi, Ghana. The study has been well conducted and reported but I still have some concerns with the self-reporting of vaccination. Most of the papers cited on self-reporting had to do with healthcare workers so it may be understandable when that is used among them. But with a big proportion of participants having basic or no education, it may be necessary to justify that among this group this kind of self-reporting is reliable. There is therefore the need to address this issue even though it has been admitted as a limitation. The analysis showing that the self-reporting was reliable based on those with vaccination cards must be explained. How many people had cards? Perhaps, if that is reported it will give more strength to the study.

Response: We thank the reviewer for this important methodological concern. We agree that justification of self-reported vaccination status is particularly necessary given the educational profile of the study population.

We now clarify that participants were asked to present vaccination cards where available, and self-reported vaccination status was cross-checked among those who possessed vaccination cards. The proportion of participants who presented vaccination cards were 224 (82.1%) of those who reported having taken the shot. This has now been reported in the manuscript to strengthen transparency and methodological rigor (See page 21, lines 391 - 393)

We further acknowledge that although self-reporting has been widely used in vaccination studies, including among non–healthcare populations, some degree of recall bias may persist, particularly among participants with lower educational attainment. This is true among the vaccinated participants who could not present their vaccination card

2. On page 18, the discussions there must be placed in context. The study population was not healthcare workers. The use of references suggesting trends like healthcare workers must be done with caution. An example, “Our finding aligns with other African studies. For example, among Ghanaian health facility staff, age group, marital status, and profession were all significantly associated with vaccination (p < 0.001) [11]. In Kampala, Uganda, lower education, age and marital status predicted greater vaccine hesitancy [41].” It must be clear in the discussion that though they are two different populations, a similar trend exists.

Response: We appreciate this important clarification. The Discussion section has been revised to explicitly distinguish the informal sector workers in the current study from healthcare worker populations cited in previous studies (see page 19, line 342 of the unmarked manuscript).

3. All numbers and percentages must be written as 515 (63.7%); and there must be a space between the number and the parenthesis. It is important that whenever a percentage is written, the number must precede it.

Response: We thank the reviewer for this stylistic correction. The entire manuscript has been carefully reviewed, and all numbers and percentages emanating from the analysis of the current study data have been reformatted to ensure that: the absolute number precedes the percentage, and a space is included between the number and the parenthesis, in accordance with journal style.

4. The authors must use data from the results sparing in the discussion. For example, “Among the unvaccinated, when asked the reason for not vaccinating, 69.0% cited cost of HBV screening and vaccination, 22.6% mentioned they do not have any reason, 16.0% cited busy schedule and 3.4% mentioned they do not know about the vaccine (Figure 2B). Meanwhile, when asked if one will take the vaccine if given free of charge, nearly 80% (79.6%) responded “Yes” and about one in five (20.4%) responded “No” (Figure 2C). “All these could have been expressed succinctly to make the point that vaccine coverage may improve if offered freely.

Response: We agree with this observation and have revised the Discussion section to summarize findings rather than restate numerical results already presented in the figures. The revised text now emphasizes interpretation - particularly the role of cost as a major barrier and the potential impact of free vaccination - without repeating percentages or detailed breakdowns (see page 20, lines 367 – 373).

5. The authors have a few typos and grammatical issues to address.

Response: We appreciate this comment. The manuscript has undergone thorough proofreading, and all identified typographical and grammatical errors have been corrected to improve clarity and readability.

We are grateful to the Academic Editor and the Reviewers for the constructive guidance, which has improved the clarity, rigor, and transparency of our manuscript. We believe that the revised version adequately addresses all concerns raised and now meets PLOS ONE’s publication criteria. We respectfully submit the revised manuscript for further consideration. Thank you.

---

## [Decision Letter · Decision Letter 1]

11 Mar 2026

Dear Dr. Obeng,

We look forward to receiving your revised manuscript.

Kind regards,

Jason T. Blackard, PhD

Academic Editor

PLOS One

Journal Requirements:

Additional Editor Comments (if provided):

Please address the remaining comments raised by the reviewers prior to acceptance.

Reviewers' comments:

Reviewer's Responses to Questions

**Comments to the Author**

Reviewer #1: (No Response)

Reviewer #2: All comments have been addressed

2. Is the manuscript technically sound, and do the data support the conclusions?

Reviewer #1: Partly

Reviewer #2: Yes

3. Has the statistical analysis been performed appropriately and rigorously?

Reviewer #1: Yes

Reviewer #2: Yes

4. Have the authors made all data underlying the findings in their manuscript fully available?

Reviewer #1: Yes

Reviewer #2: Yes

5. Is the manuscript presented in an intelligible fashion and written in standard English?

Reviewer #1: Yes

Reviewer #2: Yes

Reviewer #1: The authors have addressed most of the concerns raised satisfactorily however, these outstanding concerns need to be addressed.

In Lines 159 -164, though the authors attempted to describe how the sampling was done, it still lacks enough detail. For instance, how many occupational subgroups were sampled, what were the sizes of these subgroups, how many were sampled from each subgroup to make up the sample size of 809 stated?

Discussion - lines 328- 336 - when the results on vaccination coverage were being compared to other population groups, it would be helpful if the authors could provide some possible reasons for the similarities/differences observed compared to the other studies especially where HCWs were concerned.

Figure labels - The axes of the figures need to be labelled well. For instance, the x-axis of Figure 1A could be labelled as Hepatitis B vaccination status and 1B could be Number of doses of Hepatitis B vaccine received

Reviewer #2: The manuscript has been revised and could be published. However, the referencing seems problematic in some parts of the manuscript. I think that is a decision of the journal.

.

Reviewer #1: No

Reviewer #2: No

---

## [Author Response · Author response to Decision Letter 2]

15 Mar 2026

Manuscript ID: PONE-D-25-51851R1

Title: Towards hepatitis B elimination in Ghana: vaccination coverage and its predictors among informal sector workers in Kejetia, Kumasi, Ghana

We sincerely thank the Academic Editor and reviewers for their constructive feedback, which has helped us improve the quality and clarity of the manuscript. We have carefully revised the manuscript and addressed all comments raised. All changes have been highlighted in the tracked version of the manuscript.

Below we provide point-by-point responses.

Reviewer #1

The authors have addressed most of the concerns raised satisfactorily however, these outstanding concerns need to be addressed.

Comment 1: In Lines 159 -164, though the authors attempted to describe how the sampling was done, it still lacks enough detail. For instance, how many occupational subgroups were sampled, what were the sizes of these subgroups, how many were sampled from each subgroup to make up the sample size of 809 stated?

Response: We thank the reviewer for this important observation. The sampling procedure section has now been expanded to provide clearer detail on the occupational subgroups included in the stratified sampling approach (See page 9, line 167-169, of the unmarked manuscript).

Comment 2: Discussion - lines 328- 336 - when the results on vaccination coverage were being compared to other population groups, it would be helpful if the authors could provide some possible reasons for the similarities/differences observed compared to the other studies especially where HCWs were concerned.

Response: We appreciate this suggestion. The discussion has been expanded to include possible explanations for the differences observed between informal sector workers and healthcare workers in terms of vaccination coverage.

We now explain that healthcare workers typically have greater occupational risk awareness, institutional vaccination policies, and easier access to vaccination programs, which likely contributes to their higher vaccination coverage compared with informal sector workers (See page 20, lines 342-349 of the unmarked manuscript).

Comment 3: Figure labels - The axes of the figures need to be labelled well. For instance, the x-axis of Figure 1A could be labelled as Hepatitis B vaccination status and 1B could be Number of doses of Hepatitis B vaccine received

Response: We thank the reviewer for this observation. The figure labels have been revised to improve clarity.

Reviewer #2

The manuscript has been revised and could be published. However, the referencing seems problematic in some parts of the manuscript. I think that is a decision of the journal.

Response: We thank the reviewer for this observation. The reference list has been carefully reviewed and corrected where necessary.

We are grateful to the Academic Editor and the Reviewers for the constructive guidance, which has improved the clarity, rigor, and transparency of our manuscript. We believe that the revised version adequately addresses all concerns raised and now meets PLOS ONE’s publication criteria. We respectfully submit the revised manuscript for further consideration. Thank you.

---

## [Editor Report · Decision Letter 2]

17 Mar 2026

Towards hepatitis B elimination in Ghana: vaccination coverage and its predictors among informal sector workers in Kejetia, Kumasi, Ghana

PONE-D-25-51851R2

Dear Dr. Obeng,

We’re pleased to inform you that your manuscript has been judged scientifically suitable for publication and will be formally accepted for publication once it meets all outstanding technical requirements.

Kind regards,

Jason T. Blackard, PhD

Academic Editor

PLOS One

Additional Editor Comments (optional):

None
---

## [Editor Report · Acceptance letter]

PONE-D-25-51851R2

PLOS One

Dear Dr. Obeng,

I'm pleased to inform you that your manuscript has been deemed suitable for publication in PLOS One. Congratulations! Your manuscript is now being handed over to our production team.

Kind regards,

on behalf of

Dr. Jason T. Blackard

Academic Editor

PLOS One